# Phage Display Detection of Mimotopes that Are Shared Epitopes of Clinically and Epidemiologically Relevant Enterobacteria

**DOI:** 10.3390/microorganisms8050780

**Published:** 2020-05-22

**Authors:** Armando Navarro, Delia Licona-Moreno, Alejandro Monsalvo-Reyes, Ulises Hernández-Chiñas, Carlos A. Eslava-Campos

**Affiliations:** 1Departamento de Salud Pública, Facultad de Medicina, Universidad Nacional Autónoma de México, México City CP 04510, Mexico; liconam@unam.mx (D.L.-M.); ulisesh@unam.mx (U.H.-C.); carlos_01eslava@yahoo.com.mx (C.A.E.-C.); 2Laboratorio de Bioquímica Molecular, Unidad de Biología y Prototipos, UBIPRO, Facultad de Estudios Superiores Iztacala, Universidad Nacional Autónoma de México, Los Reyes Ixtacala Tlalnepantla, Estado de México P C 54090, Mexico; reyesac2001@gmail.com; 3Unidad Periférica de Investigación Básica y Clínica en Enfermedades Infecciosas, División de Investigación de la Facultad de Medicina, UNAM: Laboratorio de Patogenicidad Bacteriana, Unidad de Hemato-Oncología e Investigación, Hospital Infantil de México Federico Gómez/Facultad de Medicina, UNAM. Dr. Márquez 162, Col. Doctores, México City CP 06720, Mexico

**Keywords:** mimotopes, epitopes, *Salmonella*, *E. coli*, lipopolysaccharide, phagotopes, immunogenic peptides, synthetic peptides, phage display

## Abstract

Background: *Escherichia coli* and *Salmonella* are etiologic agents of intestinal infections. A previous study showed the presence of shared epitopes between lipopolysaccharides (LPSs) of *E. coli* O157 and *Salmonella*. Aim: Using phage display, the aim of this study is to identify mimotopes of shared epitopes in different enterobacterial LPSs. Methods: We use anti-LPS IgG from *E. coli* O157 and *Salmonella* to select peptide mimotopes of the M13 phage. The amino acid sequence of the mimotopes is used to synthesize peptides, which are in turn used to immunize rabbits. The antibody response of the resulting sera against the LPSs and synthetic peptides (SPs) is analyzed by ELISA and by Western blot assays, indicating that LPS sites are recognized by the same antibody. In a complementary test, the reactions of human serum samples obtained from the general population against the SPs and LPSs are also analyzed. Results: From the last biopanning phase, sixty phagotopes are selected. The analysis of the peptide mimotope amino acid sequences shows that in 4 of them the S/N/A/PF motif is a common sequence. Antibodies from the sera of immunized rabbits with SP287/3, SP459/1, SP308/3, and SP073/14 react against both their own peptide and the different LPSs. The Western blot test shows a sera reaction against both the lateral chains and the cores of the LPSs. The analysis of the human sera shows a response against the SPs and LPSs. Conclusion: The designed synthetic peptides are mimotopes of LPS epitopes of *Salmonella* and *E. coli* that possess immunogenic capacity. These mimotopes could be considered for use in the design of vaccines against both enterobacteria.

## 1. Introduction

Diarrheal illnesses are an important public health problem around the world, accounting for more than 2 million deaths each year in children under 5 years of age, with those in developing countries being the most affected [1,2]. The etiology of diarrheal illness is generally associated with viruses, parasites, and bacteria such as *Salmonella*, *Shigella*, and the different *Escherichia coli* pathotypes. All of these bacteria are important microorganisms that participate in the pathogenesis of intestinal infections [3,4].

Enterobacteria are characterized by the cytoplasmic internal membrane, peptidoglycan, and the cell wall or external membrane, which is composed of lipopolysaccharides (LPSs) and different proteins [5]. Structurally, LPSs comprise three regions. The first presents hydrophobic characteristics that include lipid A or endotoxin. The second region is the central region or the core oligosaccharides (core OSs), while the third region is known as somatic antigen O. The O antigen is a region that provides a hydrophilic surface to the bacterial cell wall and is composed of lateral carbohydrate chains that vary in their composition. This makes the O antigen useful in identifying bacteria [6,7].

Phage display, as described by Smith [8], is a procedure that uses the filamentous bacteriophage M13, in which short peptide sequences are inserted into the gene that codes for protein III (pIII), located on the phage surface [9,10]. The peptides that are expressed at random in pIII of the filamentous phage can be captured using IgG antibodies directed against specific antigens. Phage capture is based on affinity and is carried out during rounds of peptide selection, referred to as biopanning. Phages that carry peptides that mimic epitopes are known as phagotopes and the peptides that are captured by IgG are called mimotopes of the antigen, which can be used for immunization in order to produce antibodies. The phage display procedure has identified LPS mimotopes of different enterobacteria with immunogenic properties [11,12]. The immunogenicity of mimotopes has been demonstrated by inoculating them into animal models and observing how the immune response protects the animals against the enterobacteria with which the animals were challenged [11,12].

*Salmonella* is an important clinical and epidemiologic microorganism that can affect both animals and humans. The illnesses that occur in infants and adults are gastroenteritis and systemic infections, such as typhoid fever related to the *S. typhi* serotype [2,13]. On a global level, infections caused by *Salmonella* are increasing in prevalence. In the United States of America in 2011, 1.2 million cases were reported, with 450 related deaths [14]; however, there was a decline in 2013, with 19,056 cases and 80 related deaths, giving an incidence rate of 15.19 per 100,000 inhabitants [15]. In 2018, the General Epidemiology Directorate in Mexico reported 79,203 cases of *Salmonella* infection affecting all age groups [16]. Studies looking at the prevalence of *Salmonella* serotypes isolated from infected patients in both the public health sector and private hospitals showed that there were 199 serotypes involved. Of those, the most frequently occurring serotypes were *S. typhimurium*, *Salmonella* serogroup B, and *S. urbana* [17].

Infections by *E. coli* O157 are also both clinically and epidemiologically important in many countries. This microorganism has a natural reservoir in bovines, from which it is transmitted to humans by various routes [18]. *E. coli* O157 has been associated with the etiopathogenesis of hemorrhagic colitis (HC) without fever, hemolytic–uremic syndrome (HUS), and thrombocytopenic purpura [19]. The causal strains of these conditions are able to produce one or more cytotoxins. *E. coli* O157:H7 is considered to be an emerging pathogen responsible for HUS outbreaks in the United States of America, Canada, Japan, and some countries in the European Union, although the country with the highest incidence of illness caused by this bacterium is Argentina [19,20,21,22]. In a previous study using serum samples obtained from rabbits against *E. coli* O157, *S. urbana*, and *S. arizonae* LPSs, and via performing absorption assays with homologous and heterologous antigens, it was demonstrated that the presence of common epitopes exists among the mentioned LPSs [23]. In addition to the above, an ELISA test and serial dilutions of the anti-O157 LPS serum (1:50 to 1:1600) were used to analyze the reactivity against the purified LPSs from *E. coli* O157, *S. urbana*, and *S. arizonae*. By the statistical analysis of the observed reactivity (A_405_) of this serum against the homologous O157 LPS and by comparing those obtained against the LPSs from *S. urbana* and *S. arizonae*, no significant differences (*p* > 0.05) were observed. A similar assay was carried out, where the reactivity of the anti-LPS O157 serum was evaluated against the *E. coli* O179 LPS [24] and significative differences were found, showing that *E. coli* O157 and O179 LPS do not share common epitopes. The presence of immunodominant epitopes in the LPSs of *E. coli* O157 shared with the antigenic varieties of *Salmonella urbana* and *S. arizonae* [23] is interesting, since if they could be identified then they could be proposed as alternative immunogens for a wide range of clinical and epidemiologically relevant enterobacteria without the toxic effects of the LPS endotoxin (lipid A). In this study, peptide mimotopes of shared epitopes between *Salmonella* spp. and *E. coli* O157 are shown, which have been identified by the phage display procedure. The results will offer insights as to whether synthetic peptides derived from the amino acid sequence of mimotopes could be used as immunogens with the capacity to induce a protective immune response against *Salmonella* spp. and *E. coli* O157.

## 2. Materials and Methods

### 2.1. Lipopolysaccharides

We used the phenol–water method described by Westphal [25] to extract and purify the LPSs and cores from *E. coli* and *Salmonella*. The strains of *E. coli* that were used to obtain the LPSs and cores were as follows: O157 ATCC 700927, coded in the strain collection of the laboratory of the Faculty of Medicine UNAM (FMU) as 287 (FMU287); the cores were derived from mutant strains of *E. coli*, R1 (O8:K27, F470), R2 (O100:K?B:H2, F632), R3 (O111:K58:H-, F653), R4 (O14:K7, F2513), and K12 (O16, W3110) [26]; the strains of *Salmonella*, including the following: *S. urbana* (FMU459), *S. arizonae* (FMU308), and *S. typhi* ATCC 6539 (FMU073), which also came from the Faculty of Medicine, UNAM; the core Ra came from *S. minnesota* (List Biological Laboratories, Campbell, C A, USA). The LPSs and cores were treated with DNase, RNase, and proteinase K. Following extraction, the LPSs were lyophilized (Labconco, Kansas City, MI, USA) and kept refrigerated until use.

### 2.2. Anti-Sera Production Against LPS

The protocol for the research here was approved by the Research Committee (IN216417) and CONBIETICA 09CEI066201402012 of the Faculty of Medicine at the National Autonomous University of Mexico. The immunization and bleeding of rabbits was conducted in accordance with specific techniques for the production, care, and use of laboratory animals, as described in the Mexican Official Norm 062-Zoo-1999 (NOM-062-Zoo-1999; NORMA Oficial Mexicana, 1999) [27].

Before immunizing New Zealand (NZ) rabbits weighing between 2 kg and 2.5 kg, a blood sample was taken to act as a negative control for the immune reaction for the various proposed immune tests. The rabbits were obtained from the Central Animal House of the Faculty of Medicine. The rabbits were immunized with the LPSs from *E. coli* O157, *S. urbana*, *S. arizonae*, and *S. typhi* using the protocol proposed by Ewing [28]. In line with this protocol, the first dose consisted of an intradermal injection of 100 μg/mL of LPS in PBS with Freund’s adjuvant, followed by four doses in intervals of seven days. A week following the last dose, a blood sample (5 mL) was taken from the marginal vein of the rabbits and the titers against the homologous LPSs were quantified. Bloodletting was then performed on the rabbits under anesthesia. Using a ratio of 1.0 mL per kg weight, pentobarbital (Anestesal, Pfizer) was given intravenously to anesthetize the rabbits completely. The management and care of the rabbits was carried out in accordance with the Mexican official norms [27]. In accordance with the same norms, the animals were checked to be sure that they were free of transmissible diseases to humans. In addition, the blood sampling and letting was carried out according to the recommendations in Subsection 8.2 of the same Mexican official norm, which refers to the administration of fluids and substances.

### 2.3. Purification of the IgG Antibodies

IgG purification was carried out by affinity chromatography on agarose, as per the following protocol: Five-hundred microliters of sera was mixed with 250 μL of Protein G Agarose (Invitrogen) in a test tube and incubated for 20 min at an ambient temperature, which was shaken every two minutes in order to homogenize the suspension. The sample was centrifuged for 30 s at 500× *g* (Sorvall RC5B). The supernatant was eliminated and the agarose was washed 5 times with a binding buffer (0.01 M glycine with 0.15 M NaCl, pH 7.0). After the last wash, the agarose was mixed with 750 μL of an elution buffer (0.1 M glycine-HCI, pH 2.6) and centrifuged at 500× *g* for 30 s. The supernatant was collected with the IgG part and the pH was adjusted to 7.0 using 1 M Tris-HCI pH 8.0. The IgG concentration was determined by the Bradford method using the Coomassie Plus Kit (Thermo Scientific, 23236), according to the manufacturer’s instructions.

### 2.4. Biopanning and Mimotope Selection

The mimotopes were selected from a peptide library using 12 amino acid residues expressed in the filamentous phage M13 (New England BioLabs) using the method previously reported [29]. The phagotopes were selected by three rounds of selection (biopanning) with rabbit anti-LPS IgG from *E. coli* O157, *S. urbana*, *S. arizonae*, and *S. typhi*. After selecting the phagotopes, the DNA was extracted according to the Wilson method [30] and the composition of the DNA nucleotides was obtained by automated sequencing (Genetic analyzer 3100). The sequences were edited using the Chromas application and the translation of amino acids was carried out through the application of the ExPASy Proteomics Server available online http://www.expasy.ch/tools/dna.html (accessed 14-05-2017). The alignment of the amino acids and the identification of some consensual sequences were achieved by the free use of the Clustal Omega 2, available on internet (https://www.ebi.ac.uk/Tools/msa/).

### 2.5. Mimotope Synthesis

Having obtained the amino acid sequences, peptide synthesis in lineal sequences was carried out and conjugated by keyhole limpet hemocyanin (KLH) with a purity of more than 90% (Biosynthesis, Lewisville, TX, USA). For KLH-conjugated synthetic peptides (SP), an extension addition was included in the amino terminal group of amino acid residues with the sequence Cys-Ser-Gly-Gly-Gly (CSGGG).

### 2.6. Production of Anti-Sera Against Synthetic Mimoptopes by Rabbit Immunization

White New Zealand rabbits (1.5 kg) were challenged in intervals of 7 days with five doses of SP287/3, SP459/1, SP308/3, and SP073/14 conjugated with KLH. For this, 500 µg was used in the first immunization and 1.0 mg was used in the subsequent four immunizations, in accordance with the previously described method [31].

Before immunizing the rabbits, serum samples were taken for use as negative controls. One week after the final immunization, bloodletting took place and the sera were stored in aliquots at −20 °C until used. Bloodletting was carried out under anesthesia using pentobarbital (Anestesal, Pfizer). Immunization was carried out in accordance with the specific techniques for the production, care, and use of laboratory animals described in the Mexican official norms [27].

### 2.7. Evaluation of Antibody Reactivity Against LPS and Synthetic Mimotope Using ELISA

The capacity of the anti-peptide antibody reactions against SPs (SP287/3, SP459/1, SP308/3, and SP073/14) and LPSs were analyzed by the enzyme-linked immunosorbent assay (ELISA) method described by Navarro [32]. Briefly, this study used 96-well microplates (Nunc-Immuno Plate, MaxiSorp F96), into which SPs and the LPSs that were previously dissolved independently in a carbonate buffer at a pH of 9.6 were placed. The plates were incubated at 37 °C for 2 h and at 4 °C for 18–24 h, after which time the plates were blocked with 200 μL of PBS/1.0% low fat milk (Svelty, Nestlé) at ambient temperature for 2 h. The plates were washed three times with PBS/0.05% Tween 20 before the anti-peptide and anti-LPS serial dilutions (1:50 to 1:1600) of sera were dissolved in PBS at a pH of 7.4 and incubated at 37 °C for 2 h. The plates were washed a further three times (PBS/Tween 20) and 100 μL of rabbit anti-IgG (1:1000) conjugated with alkaline phosphatase (Invitrogen, USA) was added to each well. The plates were incubated at 37 °C for 2 h and the reaction was visualized by adding 200 µL of *p*-nitrophenyl phosphate (1 mg/mL, Sigma) in a diethanolamine buffer (pH 9.8, Sigma). The reaction was stopped by adding 25 µL of 3 M NaOH. An ELISA reader (BioTek EL x 800) set at 415 nm was used to read the absorbance reaction. All the tests were carried out in duplicate in two independent tests. Pre-immune rabbit sera obtained before being challenged were used as negative controls.

### 2.8. Human Serum Analysis Against LPS and Synthetic Mimotopes Using ELISA

Forty-six human serum samples from adults (age range = 15–40 years) obtained from a previous study [32] were evaluated against SP287/3, SP459/1, SP308/3, and SP073/14; and the LPSs from *E. coli* O157, *S. urbana*, *S. arizonae*, and *S. typhi*. The SPs and LPSs (10 µg/mL) dissolved in carbonate buffer (pH 9.6) were fixed in 96-well microplates (Nunc-Immuno Plate, MaxiSorp F96) and incubated at 37 °C for 2 h and at 4 °C for 18–24 h, after which time the plates were blocked with 200 μL of PBS/1.0% low fat milk (Svelty, Nestlé) at ambient temperature for 2 h.

The plates were washed three times with PBS/0.05% Tween 20 before adding human serum samples to a dilution of 1:100 in PBS at a pH of 7.4 and then incubating these at 37 °C for 2 h. The plates were washed a further three times (PBS/Tween 20) and 100 μL of human anti-IgG (1:1000) antibodies obtained in goats conjugated with alkaline phosphatase (Invitrogen, USA) was added to each well. The plates were incubated at 37 °C for 2 h and the reaction was visualized as described above. Preimmunized rabbit serum was used as negative control.

### 2.9. Interaction Site of Synthetic Antipeptide Sera

Using Western blotting, the interaction site of anti-SP287/3, anti-SP459/1, anti-SP308/3, and anti-SP073/14 sera with the LPS from *S. urbana*, *S. arizonae*, *S. typhi*, and *E. coli* O157 was analyzed, as well as the Ra, Rd, and Re cores from *Salmonella minnesota* (List Biological) and the Ra, Rb, Rc, Rd, and Re cores from *E. coli*. Using SDS-PAGE in a gel of polyacrylamide (15%) with 4 M of urea, 10 µL samples of different LPSs and cores were separated and then transferred to PVDF membranes (BioRad). Visualization was carried out using rabbit anti-IgG marked with phosphatase (Invitrogen).

### 2.10. Statistical Analysis

All ELISA tests were carried out in duplicate and the arithmetic means were compared by X^2^ proportional analysis, with a statistical significance of < 0.05, as reported previously [32].

## 3. Results

### 3.1. Mimotopes Selection

In total, 60 phagotopes were selected from the biopanning selection; 15 from each one were obtained with IgG anti-LPSs of *E. coli* O157, *S. urbana*, *S. arizonae*, and *S. typhi* LPSs. The amino acid sequences and the alignments of the 60 peptide mimotopes are presented in Appendix A. In the analysis, we observed the sequence S/N/A/PF as a consensus motif shared by the SP287/3, SP459/1, SP308/3, and SP073/14 peptides selected with IgG anti-LPSs of *E. coli* O157, *S. urbana*, *S. arizonae*, and *S. typhi*, respectively (Table 1). The composition of the consensus sequence showed serine (*S*) and asparagine (*N*) as uncharged polar amino acids and alanine (*A*) and phenylalanine (*F*) as hydrophobic amino acids. Proline (*P*), a non-polar cyclic amino acid, was present in three of the four peptide mimotopes mentioned above, while tyrosine (*Y*), an aromatic amino acid, was found in two of the peptides.

### 3.2. Anti-Peptide Sera Response

The synthesis of each peptide that presented the S/N/A/PF sequence was carried out and the resulting synthetic peptides (SPs) were used to immunize the rabbits to obtain the anti-SP287/3, SP459/1, SP308/3, and SPS073/14 sera. The anti-synthetic peptide sera as well as the anti-LPS sera were evaluated using double serial dilutions (Appendix A). The obtained results show that in all dilutions with a 1:50 dilution ratio, the anti-peptide sera reactivity is suitable (Figure 1). In this context, the ELISA test with the anti-SP287/3 serum showed OD (415 nm) lecture values of 1.65 with the homologous peptide, 0.89 with the O157 LPS, and 0.55 with the Ra core. Additionally, the serum showed reactivity with the R3 core. The same analysis of anti-SP459/1 serum reported values of 1.13 with the homologous SP, 0.62 with the *S. urbana* LPS, and 0.45 with the Ra core. This serum also recognized the R1 core (0.60). With the anti-SP308/3 serum, the obtained values were 1.19 with the homologous SP, 0.45 with the *S. arizonae* LPS, 0.51 with the Ra core, and 0.28 with R1 core. Finally, the results with anti-SP073/14 were 1.01 with the homologous SP, 0.65 with the *S. typhi* 073 LPS, and 0.55 with the Ra core. Additionally, the serum showed a response of 0.25 with the R3 core. The average values obtained with the results of the different assays with the pre-immune sera were of 0.11 OD.

### 3.3. Interaction Site of Anti-Peptide Antibodies on the LPSs

The Western blot analysis of the interaction site of anti-SP antibodies on the different LPSs showed that the anti-SP287/3 reacted with repeating carbohydrate subunits and the core region of the LPSs from *E. coli* O157. A similar response was obtained with the anti-LPS O157 antibodies (Figure 2). On the other hand, anti-SP459/1 antibody recognized a 10–15 kDa fraction corresponding to the core region and lipid A from *S. urbana* and the R1 core of *E. coli* (Figure 3). Anti-SP308/3 serum reacted with the R1 core and a 10 kDa region of the LPS from *S. arizonae (*Figure 4). Finally, anti-SP073/14 reacted with a 10 kDa region of the LPS from *S. typhi* and the R3 core of *E. coli* (Figure 5).

### 3.4. Sera Response of General Population Against Synthetic Peptides and LPSs

The ELISA assay with different serum samples obtained from the general population tested against SP287/3, SP459/1, SP308/3, and SP073/14 showed reactivity between 17% and 28% of the samples (Figure 6); however, no significant differences (*p* > 0.05) were found in any of the cases. The analysis evaluating the LPSs showed reactivity in 43.5%, 32.6%, 37.0%, and 23.9% against *E. coli* O157, *S. urbana*, *S. arizonae*, and *S. typhi*, respectively (Figure 6). The statistical analyses did not show significant differences.

## 4. Discussion

Despite the fact that mortality by intestinal infections has decreased in the general population, and more specifically in children, these infections continue to be a public health problem, representing increased morbidity [3]. Viruses are the main etiological agent of intestinal infections; however, bacteria participate in an important area of the disease. Of these infections, the enterobacteria appear with greater frequency in the pathogenesis of disease. *E. coli* O157 and *Salmonella* are bacteria that have greater clinical and epidemiologic impacts and are responsible for diarrhea outbreaks caused by consumption of contaminated food [33]. To date, there are no effective vaccines that reduce the number of people susceptible to infection by these bacteria. For this reason, it is necessary to identify specific immunogens that could protect susceptible populations.

Phage display is a versatile procedure that expresses peptides at random in protein III of the M13 phage. Some of these peptides may be mimotopes of epitopes of polysaccharides, proteins, and other molecules from microorganisms that could be used as potential immunogens in the development of vaccines [34,35].

The lipopolysaccharides of *E. coli* and *Salmonella* feature a complex structure of three regions, where the O antigen, which is constituted by repeating carbohydrate subunits, provides antigenic variability to the LPSs. The core OS, although having less variability, is also constituted by carbohydrates [6]. *E. coli* presents different core OSs, described as K12, R1, R2, R3, and R4 [26]. Additionally, for *Salmonella*, these have been described as different OS cores named Arizonae IIIA and Ra-type LPS with the chemotypes Ra, Rb, Rc, Rd, and Re. Of all of these, Ra is more commonly identified in *Salmonella* strains isolated from clinical samples [36,37,38,39]. In the present study, we utilized the phage display procedure to identify immunodominant sites present on the lipopolysaccharides of *S. typhi*, *S. urbana*, *S. arizonae*, and *E. coli* O157. For the epitope detection, we utilized IgG antibodies of immunized rabbits with the LPSs of the aforementioned bacteria. Finally, with the anti-LPS antibodies, we selected the phages expressing the peptide mimotopes. The analysis of peptide mimotopes showed the consensual sequence of S/N/A/PF in four of them. Serine, asparagine, alanine, phenylalanine, and proline residues formed a common motif in the four selected mimotopes, where each one was obtained with the different IgG anti-LPSs that were utilized. The characteristics of the motif amino acids are important, such that proline is a non-polar cyclic amino acid, where its α-amine group binds to a side chain, resulting in the formation of rigid turns into the peptide chain and causing the formation of β folds in proteins [40]. The presence of proline in the peptide mimotopes influences in the formation of curvatures in these, which could be important in the recognition of such peptide mimotopes by antibodies. Two of the selected peptides that were later synthesized (SP287/3 and SP308/3) show proline forming with the phenylalanine dipeptides (PF), a configuration that has been identified in the mimotope group A *Streptococcus* cell wall, *Vibrio cholerae* O1 Ogawa LPS, and also that of the *E. coli* O157:H7 LPS [24,41,42,43]. All these observations enables us to understand why there are cross-reactions between microorganisms of distinct genera and species, and to underline the feasibility of using mimotopes as potential immunogens.

However, James and Shin [44,45] mention that an excess of proline in the mimotopes can lead to the production of autoantibodies, so they suggest that such a situation may favor the creation of unstable configurations, and thereby the presence of multiple epitopes. However, in our study, we did not find an excess of proline in the selected peptides. It is important to carry out further studies to discard the idea that proline could have a damaging effect if used as an immunogen. Regarding other amino acids identified in the peptide mimotopes (F and Y), previous reports suggest that phenylalanine in conjunction with tyrosine and tryptophan (W) mimics structures formed by carbohydrates [41,46]. Recently, Shin [45] found phenylalanine to be an important component in 23 peptide mimotopes of the capsular polysaccharide of *Streptococcus pneumoniae*. In his study, Shin observed that the presence of this amino acid formed a dipeptide with lysin (KF). Recently, Smith [47] reported that immunized rats with peptides containing the KF dimer were protected when challenged with *S. pneumoniae*. Meanwhile, Thomas [48] identified mimotopes of lipid A in *E. coli*. In the current study, a dipeptide formed by proline and phenylalanine (PF) was identified in the peptides 287/3 and 308/3. This could be the explanation for the anti-SP287/3 and anti-SP308/3 sera response against the LPSs.

In addition to the PF dimer, the SP287/3 peptide presents another motif comprised of YxY amino acids. In a study by Oldenburg and Westerink [46,49], it was shown that the amino acid motif YxY is a carbohydrate mimotope and that the antibodies generated against the mimotope recognize concanavalin A and polysaccharides of the *Neisseria meningitidis* capsule. The presence of the YxY amino acid motif in the SP287/3 peptide suggests that the production of antibodies against this motif could be an explanation of the ability of the anti-SP287/3 serum to react against the lateral chains and core OSs of LPSs. Serine was another amino acid present in the peptide mimotopes SP287/3, SP073/14, and SP459/1. This is relevant since serine, in conjunction with proline, phenylalanine, and tyrosine, has been identified in mimotopes of carbohydrate structures of the cell wall from various microorganisms [50]. Some other dipeptides integrated by asparagine–isoleucine (NI) and isoleucine–threonine (IT) were also identified in the SP073/14 and SP459/1 peptides. In a study by Wu [51], NI and IT dipeptides were identified in the mimotope EIFTNIT of the *Streptococcus agalactiae* and *Neisseria meningitidis* capsules. The immunization of animals with the motif induced antibody production against the capsule of these microorganisms, and also against of other Gram-positive bacteria. We considered that the presence of NI and IT dipeptides in SP073/14 and SP459/1 stimulates antibody production against the LPSs evaluated in the study. In order to know the possible interaction site in the LPSs of the anti-LPS and anti-SP antibodies, we used a Western blotting assay. The results showed that each one of the different sera recognize specific LPS sites. It was observed that the anti-SP287/3 serum obtained with the corresponding SP that was selected with the anti-LPS O157 antibodies reacted with both the repeating carbohydrate subunits of the LPS and the R3 core of *E. coli* O157. Similar observations were previously reported in our laboratory [24]. In the study, it was also was observed that the anti-SP073/14 serum obtained from rabbits immunized with the SP selected with the anti-*S. typhi* LPS antibodies recognized lipid A of the *S. typhi* LPS and the R3 core of *E. coli*. In this regard, it is important to mention that the K12 and R2 cores of *E. coli* and the Ra core of *Salmonella* are conserved between both bacteria, showing a similar carbohydrate composition [52]. The existence of common epitopes between the *E. coli* and *Salmonella* LPSs was confirmed in the ELISA assay when the anti-SP287/3, anti-SP459/1, anti-SP308/3, and anti-SP073/14 antibodies were analyzed against the Ra core of *E. coli*. A possible explanation for this aspect is the fact that the Ra core was obtained from the *E. coli* EH100 strain, which presents the R2 core similarly to the *Salmonella enterica* Ra core [6,53].

To understand if previous observations occur in the human population, a collection of sera obtained previously was evaluated against the designed SPs and the LPSs of *E. coli* O157, *S. urbana*, *S. arizonae*, and *S. typhi*. The results showed that some of the population sera reacted with both SPs and LPSs, as was observed in a previous study [23]. The existence of shared epitopes that are present in the LPSs of different enterobacteria can explain the presence of antibodies against non-common bacteria, such as *E. coli* O157, as was reported in Mexico. Another interesting result observed in the study is that the peptide mimotope SP073/14 was selected with the anti-LPS sera of *S. typhi* for the serum response of anti-SP073/14 against both the SP and the *S. typhi* LPS, which suggests the potential of this peptide as a vaccine to protect against the infection of this important clinical and epidemiological bacteria. The study’s conclusion is that the response of antibodies against *S. typhi*, *S. urbana*, *S. arizonae*, and *E. coli* O157 LPS obtained from immunized rabbits with SP287/3, SP459/1, SP308/3, and SP073/14 confirms that synthetic peptides are immunogenic mimotopes of the evaluated LPSs and that it is feasible to consider them as an alternative system to protect against infections caused by *Salmonella* and *E. coli* O157; however, further studies with animal models are necessary to confirm their protective capacity.

## Figures and Tables

**Figure 1 microorganisms-08-00780-f001:**
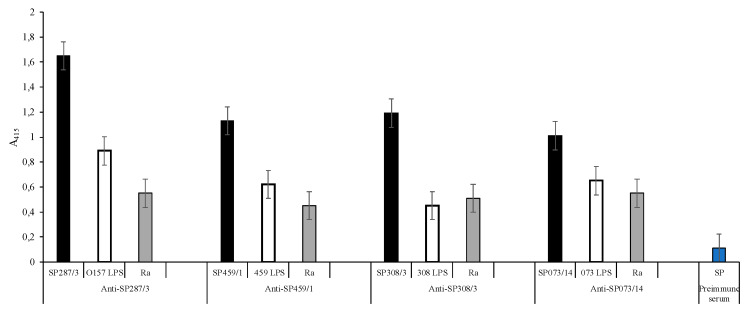
**Anti-synthetic peptide sera response.** Reactivity of sera anti-SP287/3, anti-SP459/1, anti-SP308/3, and anti-SP073/14 against synthetic peptides (
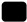
); *E. coli* O157, *S. urbana*, *S. arizonae*, and *S. typhi* LPSs (
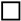
); and the Ra core (
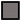
) was evaluated by the ELISA test, as was previously mentioned in the Methods section. The pre-immune sera (
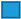
) response is the average of the obtained results in the different assays.

**Figure 2 microorganisms-08-00780-f002:**
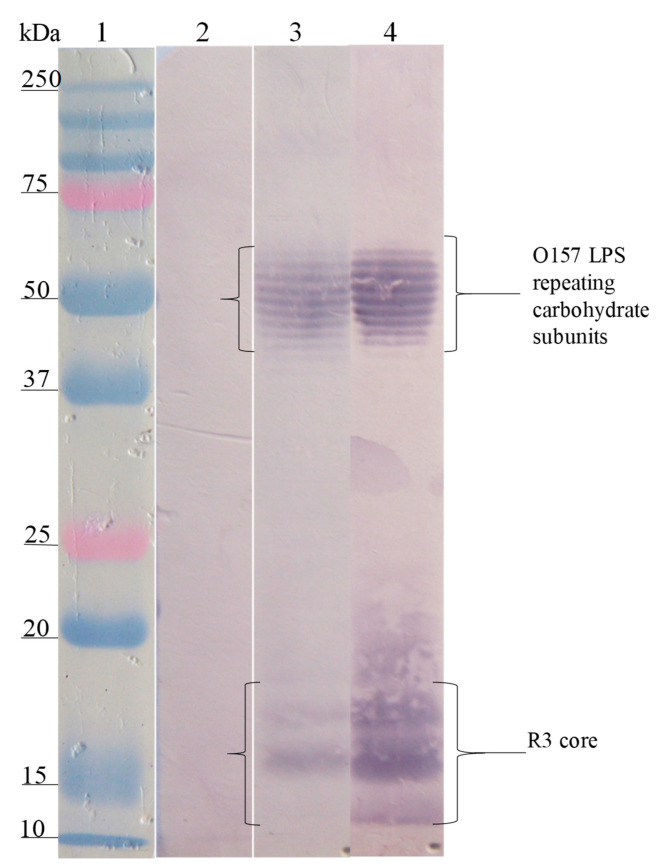
Recognition of the O157 LPS (FMU287) and R3 core by the anti-O157 LPS and the anti-SP 287/3 antibodies. Western blotting was carried out using a purified O157 LPS and R3 core. Lane 1, molecular weight marker; lane 2, O157 LPS revealed with pre-immune serum; lane 3, O157 LPS revealed with anti-O157 LPS serum; lane 4, O157 LPS revealed with anti-SP287/3 serum. The anti-O157 LPS and anti-SP287/3 antibodies recognized fractions of repeating carbohydrate subunits (stairway pattern) and the polysaccharide of the R3 core of the O157 LPS.

**Figure 3 microorganisms-08-00780-f003:**
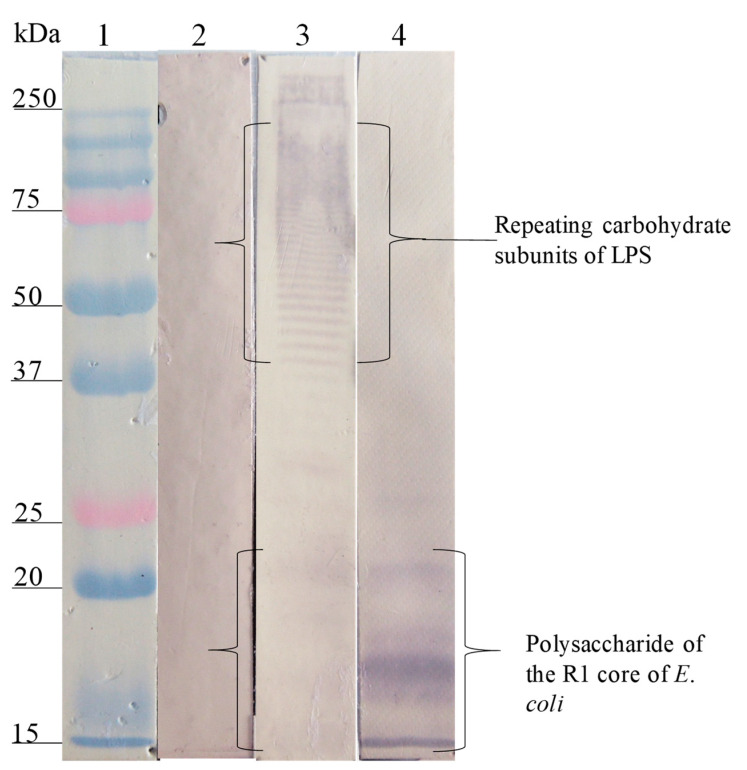
Recognition of the LPS of *S. urbana* (FMU459) and SP459/1 by the anti-LPS of *S. urbana* and the anti-SP459/1 antibodies. Western blotting was carried out using a purified LPS of *S. urbana* and the R1 core of *E. coli*. Lane 1, molecular weight marker; lane 2, LPS of *S. urbana* revealed with pre-immune antibodies; lane 3, LPS *S. urbana* revealed with anti-LPS *S. urbana* serum; lane 4, R1 core of *E. coli* revealed with anti-SP459/1 serum. The anti-LPS *S. urbana* antibodies recognized fractions of repeating carbohydrates subunits and of the core of LPS *S. urbana*. The anti-SP459/1 antibodies only recognized the R1 core of *E. coli*.

**Figure 4 microorganisms-08-00780-f004:**
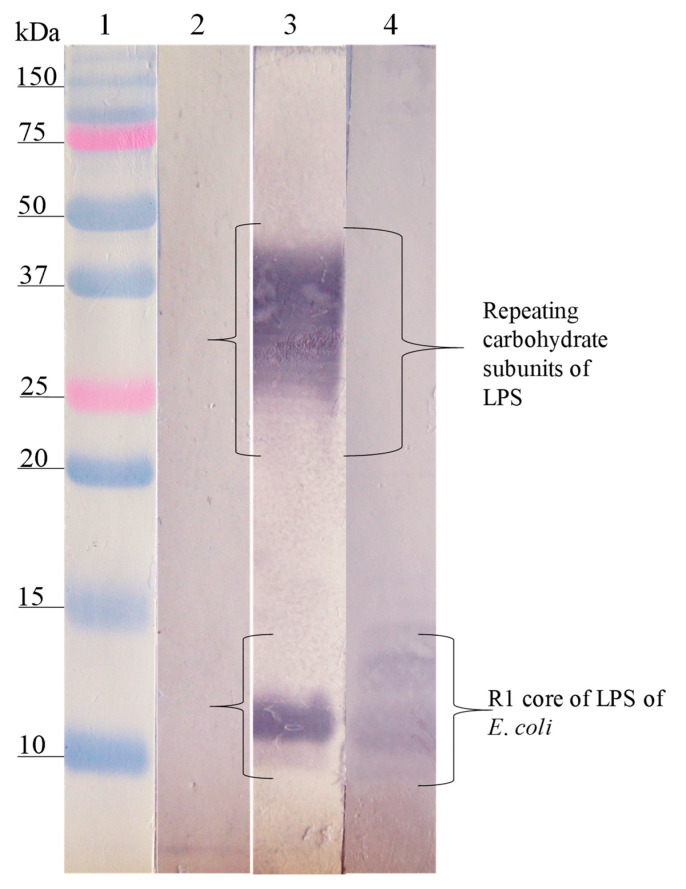
Recognition of the LPS of *S. arizonae* (FMU308) and of the R1 core of *E. coli* by the anti-LPS antibodies of *S. arizonae* and anti-SP308/3. Western blotting was carried out using a purified LPS of *S. arizonae* and the R1 core. Lane 1, molecular weight marker; lane 2, LPS of *S. arizonae* revealed with pre-immune serum; lane 3, LPS of *S. arizonae* revealed with the anti-LPS of *S. arizonae* serum; lane 4, R1 core revealed with anti-SP308/3 antibodies. The anti-LPS of *S. arizonae* antibodies recognized fractions of repeating carbohydrate subunits and the core of the LPS of *S. arizonae*. The anti-SP308/3 antibodies recognized the polysaccharide of the R1 core of *E. coli*.

**Figure 5 microorganisms-08-00780-f005:**
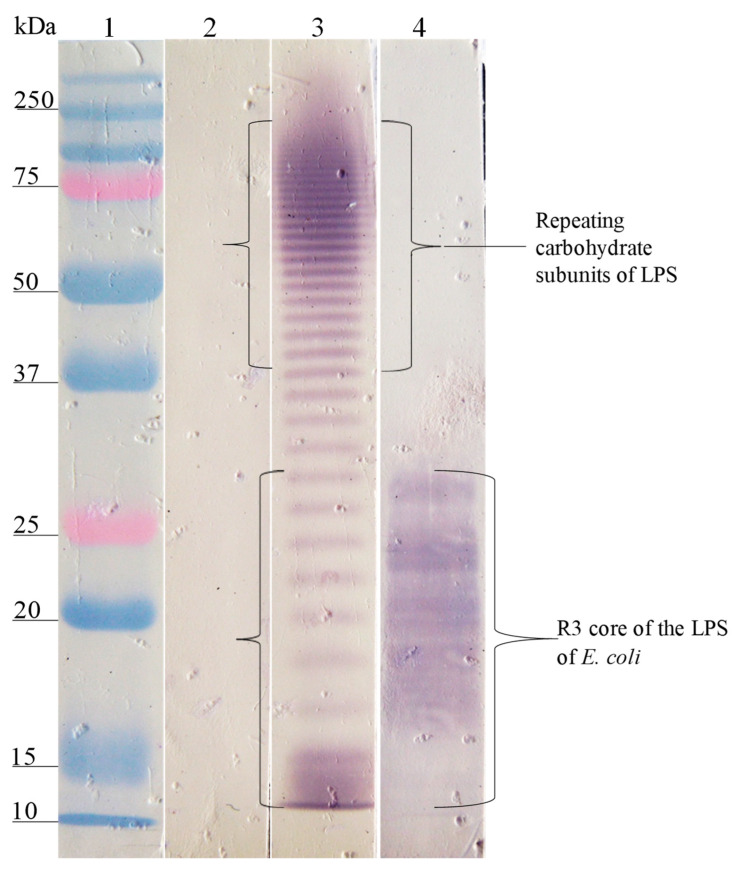
Recognition of the LPS of *S. typhi* (FMU073) and the R3 core of *E. coli* by the anti-LPS antibodies of *S. typhi* and anti-SP073/14. Western blotting was carried out using a purified LPS of *S. typhi* and the R3 core of *E. coli*. Lane 1, molecular weight marker; lane 2, LPS of *S. typhi* FMU073 revealed with pre-immune serum; lane 3, LPS of *S. typhi* revealed with anti-LPS *S. typhi* antibodies; lane 4, R3 core of *E. coli* revealed with anti-SP073/14 antibodies. The anti-LPS *S. typhi* antibodies recognized fractions of repeating carbohydrate subunits and the core of *S. typhi*. The anti-SP073/14 antibodies recognized the polysaccharide of the R3 core of *E. coli*.

**Figure 6 microorganisms-08-00780-f006:**
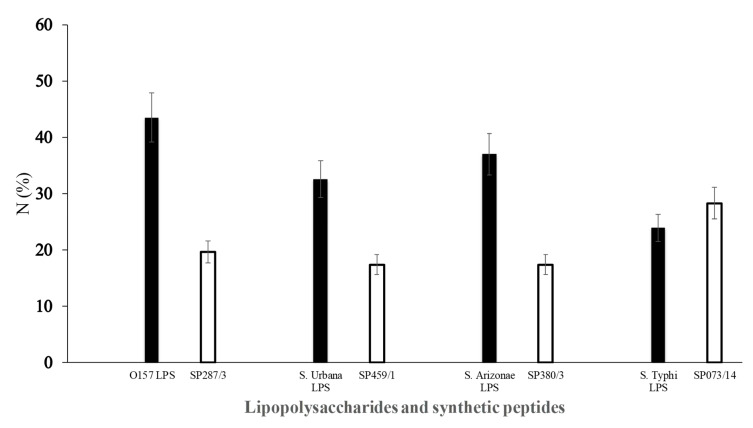
Response of human serum samples (1:100) against lipopolysaccharides and synthetic peptides. The LPSs (
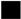
) of *E. coli* O157 (FMU287), *S. urbana* (FMU459), *S. arizonae* (FMU308), and *S. typhi* (FMU073), as well as the synthetic peptides (
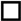
) SP287/3, SP459/1, SP308/3, and SP073/14 were analyzed independently in an ELISA test assay against community serum samples, as described previously in the Methods section.

**Table 1 microorganisms-08-00780-t001:** Alignment and consensus sequence of the selected phagotopes by phage display and anti-lipopolysaccharide (LPS) IgG response.

Phage Clones	Amino Acid Analysis and Sequence *																					Molecular Weight	Recognition by IgG anti-LPS ^1^ (A_405_ Mean)
SP287/3		S	T	L	N	Y	M	Y	X	A	H	P	F									1430.6	1.64
SP073/14	I	S	L	S	N	I	V	D	S	Q	T	P										1273.4	1.10
SP459/1			G	F	S	V	I	T	G	A	A	M	F	E								1229.4	1.4
SP308/3										H	N	P	F	T	F	F	G	P	M	F	Y	1504.7	0.63

^1^ IgG antibodies obtained from the antiserum of rabbits immunized with the purified LPSs of *E. coli* O157, *S. typhi*, *S. urbana*, and *S. arizonae*, respectively. * The motif of the consensus sequence in the four peptides was S/N/A/PF (https://www.ebi.ac.uk/Tools/msa/clustalo/).

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
