# Peer review of "Phage Display Detection of Mimotopes that Are Shared Epitopes of Clinically and Epidemiologically Relevant Enterobacteria"

_microorganisms, 2020, doi:10.3390/microorganisms8050780_

Round 1
Reviewer 1 Report
The manuscript entitled “Phage display detection of mimotopes that are shared 2 epitopes of clinically and epidemiologically relevant 3 enterobacteria” describes the selection of LPS mimotopes using as template 4 polyclonal rabbit IgG raised vs LPS from E. coli O157, S. Urbana, S. Arizonae and S. Typhi respectively. The authors claim to have identified a mimotope motif common for the four different LPS antigens. The polyclonal antibodies, which are not even affinity purified on the antigen, will select a highly diverse set of peptides from the phage display library. The authors apparently rely on the discovered common motif as the sole common factor hypothetically corresponding to the LPS specificity. The capacity of the mimotopes to induce LPS reactive antibodies is supposedly the proof.
The study suffers from multiple imperfections, practically at every step of the process:
- The polyclonal antibodies are not affinity purified on the antigen.
- The only criterion for selecting LPS mimotopes was the observation of a common motif which seems of low statistical significance. It would help to report the significance of the motif.
- Biopanning data should be checked for target unrelated peptides at http://i.uestc.edu.cn/bdb.
- The immunogenicity of the mimotopes is proven by the anti-LPS reactivity of the sera from mimotope immunized animals. This seems a convincing proof but the figure presents a single dilution reactivity. Moreover, the dilution is not reported. The figure should present binding assay with serial dilutions of the sera. Since the reason for selecting these mimotopes was a common motif It would be interesting to see also the cross-reactivity between the mimotopes and the target LPS antigens.
- The cross-reactivity remains an unanswered question also in the epitope identification experiments. After all, in the title the authors claim they have mimotopes of shared epitopes. Instead, the Western blot with anti-SP287/3 serum shows reactivity both with the repeating units of the O antigen as well as with the R3 core. Is this a common epitope or is this a case of multiple antigen mimotopes that have been observed previously with carbohydrate mimotopes? It would be interesting to see reactivities to an array of carbohydrate antigens to demonstrate the specificity of the induced antibodies to LPS and to the given LPS antigen or to
- The final experiment seems useless because there is no reason to believe that there is any relationship between LPS and mimotope reactivities. Again, single dilution binding assay is suboptimal and even the values of the LPS and mimotope reactivities do not seem to correlate.
Apart from these shortcomings, there are numerous technical issues with the manuscript. The English is poor and many sentences need rewording (e.g. – lines 198-202, 299-301,307 just to name a few). The legends of the figures 1 and 6 are in disarray. Also, it would help to have a column in Table 1 indicating the antigen specificity of the template.
Thus, this is a potentially interesting study which may be published only after major revision.
Author Response
Reviewer #1:
1.-The polyclonal antibodies are not affinity purified on the antigen.
Reply, in line 133: to clarify the paragraph corresponding to “Obtaining polyclonal anti-LPS IgG”, the subtitle was modified by “Purification of the IgGs antibodies”
2.- The only criterion for LPS mimotopes selection was the observation of a common motif which seems of low statistical significance. It would help to report the significance of the motif.
Reply, in lines 81-90, the following paragraph was introduced: “In a previous study using serum samples obtained from rabbits against E. coli O157, S. urbana, and S. arizonae LPSs, and via performing absorption assays with homologous and heterologous antigens, it was demonstrated that the presence of common epitopes exists among the mentioned LPSs [23]. In addition to the above, an ELISA test and serial dilutions of the anti-O157 LPS (1:50 to 1: 1600) were used to analyze the reactivity against the purified LPSs from E. coli O157, S. urbana, and S. arizonae. By the statistical analysis of the observed reactivity (A405) of this serum against the homologous O157 LPS and comparing those obtained against the LPSs from S. urbana and S. arizonae, no significant differences (p > 0.05) were observed. A similar assay was carried out, where the reactivity of the anti-LPS O157 serum was evaluated against the E. coli O179 LPS [24] and significative differences were found, showing that E. coli O157 and O179 LPS do not share common epitopes.”
3.- Biopanning data should be checked for target unrelated peptides at http://i.uestc.edu.cn/bdb.
Reply: The amino acid sequence of the mimotope peptides in the TUPScan system was reviewed, in this regard it was not found reported amino acid sequences similar to the peptides of the current study. The report is attached,
4.- The immunogenicity of the mimotopes is proven by the anti-LPS reactivity of the sera from mimotope immunized animals. This seems a convincing proof but the figure presents a single dilution reactivity. Moreover, the dilution is not reported. The figure should present binding assay with serial dilutions of the sera. Since the reason for selecting these mimotopes was a common motif It would be interesting to see also the cross-reactivity between the mimotopes and the target LPS antigens.
Reply: The results of each anti-peptide mimotopes sera reactivities were reviewed again, in this regard, a Supplementary Table 1 was prepared in which the serial titration of each antisera (1:50 to 1:1600) are presented. To analyze this the Supplementary Table 1 is attached. In the same way, with the updated review the Figure 1 was modified. Clarifying that, based on the results obtained from de titration of each antisera, the 1:50 dilution was chosen to exemplify the results of the titrations of the anti-peptide mimotope sera.
The reviewer's suggestion led to insert the following paragraph to point out that the reactivity of the anti-peptide serum was evaluated by performing serial dilutions of the sera:
Lines 237-248: The synthesis of each peptide that presented the S/N/A/PF sequence was carried out and the resulting synthetic peptides (SPs) were used to immunize the rabbits to obtain the anti-SP287/3, SP459/1, SP308/3, and SPS073/14 sera. The anti-synthetic peptide sera were evaluated using double serial dilutions (Supplementary Table 1). The obtained results show that in all dilutions with a 1:50 dilution ratio that the anti-peptide sera reactivity is suitable (Figure 1). In this context, the ELISA test with the anti-SP287/3 serum showed OD (415 nm) lecture values of 1.65 with the homologous peptide, 0.89 with the O157 LPS, and 0.55 with the Ra core. Additionally, the serum showed reactivity with the R3 core. The same analysis of anti-SP459/1 serum reported values of 1.13 with the homologous SP, 0.62 with the S. urbana LPS, and 0.45 with the Ra. This serum also recognized the R1 core (0.60). With the anti-SP308/3 serum, the obtained values were 1.19 with the homologous SP, 0.45 with the S. arizonae LPS, 0.51 with the Ra core, and 0.28 with R1 core. Finally, the results with anti-SP073/14 were 1.01 with the homologous SP, 0.65 with the S. typhi 073 LPS, and 0.55 with the Ra core. Additionally, the serum showed a response (0.25) with the R3 core. The average values obtained with the results of the different assays with the pre-immune sera were of 0.11 OD.
5.- The cross-reactivity remains an unanswered question also in the epitope identification experiments. After all, in the title the authors claim they have mimotopes of shared epitopes. Instead, the Western blot with anti-SP287/3 serum shows reactivity both with the repeating units of the O antigen as well as with the R3 core. Is this a common epitope or is this a case of multiple antigen mimotopes that have been observed previously with carbohydrate mimotopes?
Reply: The results of the reactivity of the anti-SP287/3 serum in the Western blot showed that recognition of both the side chains and the R3 core region, such recognition of the two structures was confirmed using ELISA assay, these were positive for the LPS and the R3 core.
This reactivity could be explained by common carbohydrates present in the two structures, such as glucose and galactose.
Also the R3 core was recognized in Western blot by the SP073/14 anti-serum, this recognition was confirmed by ELISA test, therefore the peptides SP278/3 and SP073/14 are mimotopes of the R3 core.
6.- It would be interesting to see reactivities to an array of carbohydrate antigens to demonstrate the specificity of the induced antibodies to LPS and to the given LPS antigen or to
Reply: A Supplementary Table 2 was prepared to show the cross antigenic reactivity of anti-sera against LPS from E. coli O157, S. Urbana and S. Arizonae against LPS and mimotope peptides.
7.-The final experiment seems useless because there is no reason to believe that there is any relationship between LPS and mimotope reactivities. Again, single dilution binding assay is suboptimal and even the values of the LPS and mimotope reactivities do not seem to correlate.
Reply: With the final experiment, we try to show the immunogenicity of synthetic peptides, the recognition of the peptides by the human serum samples showed that the synthetic peptides of the current study are immunogenic. However, we agree with the reviewer that the reactivity values do not correlate, and the above is expected since the mimotope only mimics a part of the LPS.
Apart from these shortcomings, there are numerous technical issues with the manuscript. The English is poor and many sentences need rewording (e.g. – lines 198-202, 299-301,307 just to name a few). The legends of the figures 1 and 6 are in disarray. Also, it would help to have a column in Table 1 indicating the antigen specificity of the template.
8.- Lines 198-202: “Composition of the amino acid consensus sequence showed that Serine (S) and Asparagine (N) were non-charged polar amino acids, while Alanine (A) and Phenylalanine (F) were hydrophobic amino acids. Proline (P) was shown to be a non-polar cyclic amino acid, which was present in three of the four peptide mimotopes mentioned above, and the aromatic amino acid Tyrosine (Y) was found in two of the Peptides”.
Reply: Lines 218-222 “The composition of the consensus sequence showed serine (S) and asparagine (N) as uncharged polar amino acids and alanine (A) and phenylalanine (F) as hydrophobic amino acids. Proline (P), a non-polar cyclic amino acid, was present in three of the four peptide mimotopes above mentioned, tyrosine (Y), an aromatic amino acid, was found in two of the peptides”.
9.- Lines 299-301: “Lipopolysaccharides of E. coli and Salmonella are compounds in the O antigen, which is a region integrated by repeating carbohydrate subunits that provide antigenic variability to the molecule. However, the core OS, which is also composed of carbohydrates, present less variability [6]
Reply, Lines 322-325: The lipopolysaccharides of E. coli and Salmonella feature a complex structure of three regions, where the O antigen, which is constituted by repeating carbohydrate subunits, provides antigenic variability to the LPSs. The core OS, although with less variability, also is constituted by carbohydrates [6]
10.- Line 307. “We selected phagotopes from polyvalent sera prepared in rabbits against the LPSs of the aforementioned bacteria”.
Reply, lines 330-331: “For the epitope detection, we utilized IgG antibodies of immunized rabbits with the LPSs of the aforementioned bacteria”
11.- Lines 337-340: “Oldenburg and Westerink [46, 49] reported that YxY amino acids form a mimotope of carbohydrates and that the antibodies generated in rabbits against the mimotope were joined to Concanavalin A and to polysaccharides of the Neisseria meningitidis capsule.
Reply, lines 363-365: The Phrase was changed to: In a study of Oldenburg and Westerink [46,49], it was referred that the amino acid motif YxY is a carbohydrate mimotope and that the antibodies generated against the mimotope recognize concanavalin A and polysaccharides of the Neisseria meningitidis capsule.
12.- Also, it would help to have a column in Table 1 indicating the antigen specificity of the template.
Reply: In Table 1, a column was added with the results of recognition by IgG anti-phage clone (A415 mean).
13.- The legends of the figures 1 and 6 are in disarray.
Reply :The legends of the figures were fixed. The disorganization of the Figures was due to the fact that the system of Microorganisms rearranged the type and size of the letter.
14.- Extensive editing of English language and style required
Reply: The manuscript was submitted to revision and edition of English in the MDPI system, certificate is attached.

Reviewer 2 Report
The results shown by Navarro, et al., are interesting and present the procedure of the identification of mimotopes of LPS epitopes of pathogenic enterobacteria by using M13 phage. The main conclusions are supported by the presented data. However, there are some specific points which have to be improved:
- In the chapter Introduction, the Authors wrote that the purpose of their study is to identify the epitopes shared between both Salmonella and E. coli O157 (lines 81-82). Please explain in the text, why the identification of common epitopes is so important?
- In the chapter “Selection of mimotopes” The authors wrote, that from 60 phagotopes, only 4 peptides shared S/N/A/PF consensus motif? And what about the others, did they have any common characteristics?
- Why the Authors used only coli O157 in their experiments? And what about the other serotypes that are also dangerous to humans ( for example: O11, O26, O104).
- Please create the scheme to sum up the procedure of identification of mimotopes.
- The manuscript contains many language errors. In my opinion, it should be corrected by native speaker.
Author Response
Reviewer #2:
The results shown by Navarro, et al., are interesting and present the procedure of the identification of mimotopes of LPS epitopes of pathogenic enterobacteria by using M13 phage. The main conclusions are supported by the presented data. However, there are some specific points which have to be improved:
- In the chapter Introduction, the Authors wrote that the purpose of their study is to identify the epitopes shared between both Salmonella and E. coli O157 (lines 81-82). Please explain in the text, why the identification of common epitopes is so important?
Reply, lines 93-95: The following sentence the phrase was inserted: is interesting, since if they could be identified then they could be proposed as alternative immunogens for a wide range of clinical and epidemiologically relevant enterobacteria without the toxic effects of the LPS endotoxin (lipid A).
In the chapter “Selection of mimotopes” The authors wrote, that from 60 phagotopes, only 4 peptides shared S/N/A/PF consensus motif? And what about the others, did they have any common characteristics?
Reply: An analysis of the alignment and of the amino acid sequence of the 60 phagotopes is presented in the Supplementary Table 3. Likewise, in this Table the consensus sequence was determined in each phagotope groups.
- Why the Authors used only coli O157 in their experiments? And what about the other serotypes that are also dangerous to humans (for example: O11, O26, O104).
Reply: The current study is the product of a study line of several years in which we have addressed the issue of seeking alternatives for the development of protective immunogens. In this regard, we have found that E. coli O157 LPS, presents common epitopes with E .coli from serogroups O7, O116 as well as with two of Salmonella serovarieties (S. Urbana and S. Arizonae), in such a way that this project we used Phage display to know if this methodology allowed us to know what epitopes of LPS could be the shared ones. But, we are very grateful for the observation of looking towards to other LPS of the E. coli serogroups, which we will take into account for future Projects.
- Please create the scheme to sum up the procedure of identification of mimotopes.
Reply. Scheme to sum up the procedure was prepared, a file is attached as a Supplementary Figure 1.
The manuscript contains many language errors. In my opinion, it should be corrected by native speaker.
Reply: The manuscript was submitted to English edition in the MDPI Author Services, certificate is attached.

Round 2
Reviewer 1 Report
The manuscript is publishable in th epresent form.